# Differential Infiltration of T-Cell Populations in Tumor and Liver Tissues Predicts Recurrence-Free Survival in Surgically Resected Hepatocellular Carcinoma

**DOI:** 10.3390/cancers17091548

**Published:** 2025-05-02

**Authors:** Eun Ji Jang, Ho Joong Choi, Young Kyoung You, Deok Hwa Seo, Mi Hyun Kwon, Keungmo Yang, Jaejun Lee, Jeong Won Jang, Seung Kew Yoon, Ji Won Han, Pil Soo Sung

**Affiliations:** 1Department of Biomedicine and Health Sciences, The Catholic University Liver Research Center, College of Medicine, POSTECH-Catholic Biomedical Engineering Institute, The Catholic University of Korea, Seoul 06591, Republic of Korea; dmswl9027@catholic.ac.kr (E.J.J.); smin0904@catholic.ac.kr (D.H.S.); myun1ng@catholic.ac.kr (M.H.K.); 2The Catholic University Liver Research Center, College of Medicine, Department of Biomedicine & Health Sciences, The Catholic University of Korea, Seoul 06591, Republic of Korea; yang27jin@catholic.ac.kr (K.Y.); pwln0516@gmail.com (J.L.); garden@catholic.ac.kr (J.W.J.); yoonsk@catholic.ac.kr (S.K.Y.); 3Department of Surgery, Seoul St. Mary’s Hospital, Seoul 06591, Republic of Korea; hopej0126@catholic.ac.kr (H.J.C.); yky602@catholic.ac.kr (Y.K.Y.); 4Division of Gastroenterology and Hepatology, Department of Internal Medicine, College of Medicine, Seoul St. Mary’s Hospital, The Catholic University of Korea, Seoul 06591, Republic of Korea

**Keywords:** hepatocellular carcinoma, liver-infiltrating T cells, tumor-infiltrating T cells, recurrence-free survival, resection

## Abstract

Liver cancer, or hepatocellular carcinoma (HCC), has a poor prognosis, and understanding how immune cells influence tumor progression is crucial for improving treatments. However, the roles of different T-cell types, such as CD69^+^ and CD69^−^ T cells, remain unclear, particularly in relation to their location within or outside the tumor. This study aims to clarify their functions and prognostic significance in HCC. We found that CD69^−^ T cells within tumors are linked to better patient outcomes, likely due to reduced exhaustion, while CD69^+^CD103^+^CD8^+^ T cells in non-tumor liver tissues play a key role in immune surveillance. These findings suggest that analyzing immune cells based on their spatial distribution could improve immunotherapy strategies. Additionally, these T cells may serve as biomarkers for predicting cancer recurrence and treatment response, paving the way for more personalized and effective treatment approaches.

## 1. Introduction

Hepatocellular carcinoma (HCC) is the most common primary liver cancer and the fourth leading cause of cancer-related mortality worldwide [1]. Surgical resection is the recommended treatment for early stages in patients with HCC having a single tumor. This treatment is also considered a curative option; however, a substantial proportion of these patients experience tumor recurrence during the postoperative period [2,3]. Some prediction models for tumor recurrence following surgical resection have been suggested. These models use various clinical or histopathologic factors. However, various criteria across different guidelines or clinical trials impact their use, hence highlighting the need to develop biomarkers to predict recurrence and guide future investigation of perioperative treatment strategies [4].

T cells are crucial for anti-tumor immunity and cancer therapy. CD8^+^ cytotoxic T lymphocytes (CTLs) target cancer cells by recognizing tumor-specific antigens on major histocompatibility complex (MHC) class I molecules and induce apoptosis through perforin and granzymes [5]. CD4^+^ helper T cells support this process by secreting cytokines, such as interleukin-2 and interferon-γ, which activate and expand CTLs and also recruit other immune cells [6]. Cancer cells evade T-cell responses by downregulating MHC class I or expressing inhibitory molecules like programmed death ligand 1 (PD-L1) [7,8]. Additionally, they create immunosuppressive environments that weaken T-cell functions. Immune checkpoint inhibitors, such as those targeting programmed death protein 1 (PD-1) or CTL-associated protein-4 (CTLA-4), restore T-cell activity and enhance anti-tumor immunity [9,10]. Therefore, understanding these mechanisms is vital for developing innovative therapies and improving cancer treatment outcomes [11].

Tissue-infiltrating T cells significantly contribute to modulating immune responses in the liver and the tumor microenvironment (TME) of HCC. These cells can be classified into tissue-resident memory T (T_RM_) cells, characterized by CD69 expression, and recirculating T cells (CD69^−^ T cells), which lack CD69 expression [12]. CD69^+^ T_RM_ cells are widely involved in tissue retention and local immune surveillance [13]. CD103, which is expressed on a subset of T_RM_ cells, interacts with E-cadherin to mediate adhesion to epithelial cells [14]. In contrast, CD69^−^ T cells are considered a population that contains recirculating potential; however, their characteristics compared with those of peripheral T cells or other tissue-infiltrating populations are not well understood [15,16].

Most studies on the association between tissue-infiltrating T cells and the clinical outcomes of patients with HCC focused on the T_RM_ populations. One study reported that tumor-infiltrating CD8^+^ T_RM_ cells express high levels of PD-1, and their increased infiltration was correlated with poor overall survival (OS) in HCC cases [17]. However, another study revealed that increased tumor infiltration of CD8^+^ T_RM_ cells correlated with good OS for patients [18]. These variations may be due to the heterogeneity of the tumor-infiltrating populations, suggesting the need for a detailed analysis. Furthermore, liver-infiltrating T-cell populations, which might be important in postoperative tumor immune surveillance, should be investigated as a therapeutic target or predictive biomarker.

Therefore, in the present study, we aimed to investigate the distinct characteristics of CD69^+^ and CD69^−^ T cells of the liver and tumor and their associations with clinical or histopathological parameters. Notably, we analyzed whether the frequencies of T-cell subpopulations can predict recurrence-free survival (RFS). These data may provide unique information for perioperative treatment and patient selection, considering the findings from the current IMbrave050 clinical trials using immune-checkpoint inhibitors after surgical resections in patients with HCC having a high risk of recurrence [19,20].

## 2. Materials and Methods

### 2.1. Study Samples

This study was conducted with a patient cohort established between October 2022 and August 2024. In total, 57 patients with HCC who underwent surgical resection were consecutively enrolled in this study. The study protocol was approved by the Institutional Review Board of Seoul St. Mary’s Hospital, and informed consent was obtained from all patients (approval number: KC21TISI0413). In addition, the authors have no conflicts of interest to declare. The study was performed following the principles outlined in the Declaration of Helsinki. HCC tissues (tumor) and the farthest liver tissues (non-tumor) were collected to analyze liver tissue-infiltrating lymphocytes during hepatectomy (Appendix A). Single-cell suspensions were prepared using the Tumor Dissociation Kit (Miltenyi Biotec, Bergisch Gladbach, Germany) and gentleMACS Dissociator (Miltenyi Biotec, Bergisch Gladbach, Germany) following the manufacturer’s instructions. The dissociation process included enzymatic digestion at 37 °C for 30 min with periodic mechanical agitation using a mixture of Enzyme H, Enzyme R, and Enzyme A, as provided in the Tumor Dissociation Kit (Miltenyi Biotec, Bergisch Gladbach, Germany). The resulting suspensions were filtered through a 70 µm cell strainer (Falcon, Corning Inc., Corning, NY, USA) to remove debris and undigested tissue fragments, which was followed by centrifugation using an Allegra X-15R Centrifuge (Beckman Coulter, Brea, CA, USA) at 300× *g* for 10 min. Isolated cells were either cryopreserved in a solution containing fetal bovine serum (Gibco, Thermo Fisher Scientific, Waltham, MA, USA) and 10% dimethyl sulfoxide (Sigma-Aldrich, St. Louis, MO, USA) or immediately used for further experiments. There was no significant difference in marker expression between fresh and cryopreserved samples. The diagnosis of HCC was made based on the most recent guidelines of the Korean Association for the Study of the Liver [2]. Alcohol-related liver disease (ALD) was defined as chronic alcohol consumption exceeding 60 g/day for males and 50 g/day for females, as previously described [21]. Based on this definition, a total of 21 patients with alcohol-related etiology, including those with HBV combined with ALD, were included in the analysis.

### 2.2. Flow Cytometry

Fluorescence staining was performed on isolated cells using surface marker antibodies at room temperature for 15 min. The antibodies used for flow cytometry are as follows: BV421-conjugated anti-CD69 (BioLegend, San Diego, CA, USA), V500-conjugated anti-CD3 (BD Biosciences, Franklin Lakes, NJ, USA), PerCP-Cy5.5-conjugated anti-CD4 (BD Biosciences, Franklin Lakes, NJ, USA), PE-conjugated anti-CD103 (BioLegend, San Diego, CA, USA), PE-Cy7-conjugated anti-natural killer group 2 member D (NKG2D) (BioLegend, San Diego, CA, USA), PE-Texas Red-conjugated anti-CD14 (eBioscience, San Diego, CA, USA), PE-Texas Red-conjugated anti-CD19 (eBioscience, San Diego, CA, USA), APC-conjugated anti-PD-1 (BioLegend, San Diego, CA, USA), APC-H7-conjugated anti-CD8 (BD Biosciences, Franklin Lakes, NJ, USA), and LIVE/DEAD red fluorescent reactive dye (Invitrogen, Waltham, MA, USA). Multicolor flow cytometry was conducted using the LSRFortessa flow cytometer (BD Biosciences, Franklin Lakes, NJ, USA), and the data were analyzed using FlowJo software version 10.9.0 (TreeStar, Ashland, OR, USA).

### 2.3. Statistical Analysis

Univariate Cox proportional hazards regression analyses were performed to evaluate the associations between tissue-infiltrating T-cell populations, clinicopathologic factors, and RFS. The Cox model was chosen because it appropriately accounts for time-to-event data and censoring, making it suitable for survival data analysis. Hazard ratios (HRs) and the corresponding 95% confidence intervals (CIs) were calculated to quantify the strength of these associations. Kaplan–Meier curves were used to evaluate RFS across different groups. Statistical significance was set at *p* < 0.05. Continuous variables were compared using independent *t*-tests, and relationships between two parameters were assessed using the Pearson correlation coefficient. The Mann–Whitney U test was used to compare data between two independent groups. For multiple group comparisons, statistical significance was determined using one-way analysis of variance. Furthermore, propensity score matching with one-to-one nearest-neighbor matching within a 0.20 caliper width was used to mitigate baseline differences between groups. All statistical analyses were performed using R statistical software version 4.0.3 (R Foundation for Statistical Computing, Vienna, Austria) and GraphPad Prism version 8 (GraphPad Inc., La Jolla, CA, USA).

## 3. Results

### 3.1. Differential Expressions of Tissue-Resident Markers Among T Cells Between Tumor and Background Liver Tissues

In total, we consecutively enrolled 57 patients (Table 1). These patients comprised individuals with HCC caused by hepatitis B virus (HBV) (n = 28), alcohol-related liver disease (Alc, n = 9), HBV combined with ALD (HBV + Alc, n = 12), and others (n = 8). The mean age of all patients was 62.7 years, with more men (n = 43, 75%) than women. The median alpha-ferroprotein (AFP) and protein induced by vitamin K absence or antagonist-II (PIVKA-II) levels were 4.5 ng/mL and 43.0 mAU/mL, respectively. The median size of the largest tumor was 3.7 cm, and microvascular invasion was observed in 33% of the patients (n = 19). The Cancer Genome Atlas (TCGA) liver cancer dataset was used to stratify patients into high (hi) and low (lo) expression groups based on the expression levels of CD3, CD4, and CD8 markers, and their OS was compared (Appendix A). The results showed that CD3 expression did not affect the OS, whereas CD4 and CD8 expressions were weakly associated with the OS, suggesting that detailed analysis of T-cell subsets in HCC cases may be needed to identify their associations with the patients’ outcomes. Subsequently, we analyzed the expressions of CD69 and CD103 on CD4^+^ and CD8^+^ T cells, comparing the relative frequencies of each subset between the two tissue types (Figure 1A,B). The gating strategy for the flow cytometry analysis is illustrated in Figure 1. The results of the analysis of the relative frequencies of CD4^+^ and CD8^+^ T cells between liver and tumor tissues showed that tumor tissues contained a significantly higher number of CD4^+^ T cells but lower number of CD8^+^ T cells compared to the background liver tissues, suggesting distinct infiltrations of T-cell populations among both tissue types. Regarding CD69 and CD103 expressing populations, CD69^+^ populations were the major populations among CD4^+^ and CD8^+^ T cells of both tissue types. However, CD69^+^CD4^+^ T cells barely expressed CD103 compared with CD69^+^CD8^+^ T cells, and slightly lower frequencies of CD69^+^CD8^+^ T cells were observed in the tumor tissues than in liver tissues (Figure 1B).

### 3.2. Associations Between Clinical or Pathologic Characteristics and Each T-Cell Subpopulation Within Tumor and Background Liver Tissues

Next, we investigated whether differential infiltration of liver and tumor T cells is associated with clinical or pathologic characteristics (Figure 2A). The results showed that liver inflammation reflected by the serum aspartate aminotransferase (AST) level correlated with the infiltration of CD69^+^CD103^−^ T cells of liver tissue. Additionally, the frequency of liver CD69^+^CD4^+^ T cells was associated with the serum total bilirubin level, whereas the serum albumin level negatively correlated with the frequency of liver CD69^+^CD8^+^ T cells. Furthermore, portoperiportal inflammation of the liver positively correlated with CD4^+^ T cells; however, it negatively correlated with CD8^+^ T cells. A higher tumor grade positively correlated with CD69^+^CD103^+^CD4^+^ T cells in the tumor. Notably, alcohol intake was significantly associated with the reduced infiltration of CD69^−^CD4^+^/CD8^+^ T cells both in the liver and tumor tissues.

The relative frequencies of T-cell subpopulations in liver and tumor tissues showed diverse patterns and variations depending on alcohol etiology (Figure 2B,C). In the liver tissue, the frequency of CD69^−^CD4^+^ T cells was significantly lower, whereas the frequencies of CD69^+^, CD69^+^CD103^+^, and CD69^+^CD103^−^CD4^+^ T cells were significantly higher when alcohol etiology was present than when it was not. In contrast, only the frequency of CD69^+^CD103^+^CD8^+^ T cells was significantly increased in patients with alcohol etiology. In tumor tissues, the frequency of CD69^−^CD4^+^ T cells was significantly lower, whereas those of CD69^+^ and CD69^+^CD103^−^CD4^+^ T cells were significantly higher in patients with alcohol etiology than in those without (Figure 2C), although there were no statistically significant differences observed in CD8^+^ T-cell subpopulations in tumor tissues based on alcohol etiology. These findings demonstrate the varying impact of alcohol etiology on the distribution of CD4^+^ T-cell subpopulations in both tissue types. To further support these observations, we performed a subgroup analysis limited to patients with ALD (i.e., without HBV co-infection). While the reduced sample size precluded statistical significance, the overall distributional trends were preserved. The results of this subgroup analysis are presented in Appendix A.

### 3.3. Clinical Relevance of Relative Infiltration of Each T-Cell Subpopulation Within Tumor and Background Liver Tissues in Patients with HCC with Resection

Next, we investigated whether differential infiltration of liver and tumor T cells predicts clinical outcomes in surgically resected patients with HCC. In liver tissues, increased frequencies of CD69^+^CD8^+^ and CD69^+^CD103^−^CD8^+^ T cells were associated with a high risk of recurrence (Table 2), whereas a high frequency of CD69^+^CD103^+^CD8^+^ T cells correlated with a reduced recurrence risk (*p* < 0.005, Table 2), suggesting a protective role for this subset. In contrast, within tumor tissues, T cells expressing CD69, including the CD69^+^CD103^+^CD8^+^ population, were associated with an increased recurrence risk, whereas increased frequencies of CD69^−^CD4^+^ and CD69^−^CD8^+^ T cells were associated with better outcomes (Table 2). Of the clinicopathological factors, the largest tumor size, followed by elevated PIVKA-II, AFP, and pathological T stage showed the strongest correlation with recurrence (Table 2), whereas patient age, liver function parameters, tumor number, and histological fibrosis were not significantly associated. Furthermore, the alcohol-related etiology of HCC did not influence RFS (*p* = 0.718, Appendix A). The lack of a significant difference in RFS between alcohol-related and non-alcohol-related patients with HCC (Appendix A) suggests that even though alcohol etiology influences T-cell subset distributions and phenotypes, it may not directly determine long-term survival outcomes. These findings were further supported by the Kaplan–Meier analyses, which revealed that patients with higher frequencies of CD69^+^CD103^+^CD8^+^ T cells in liver tissues (Figure 3A) experienced significantly improved RFS, whereas those with higher frequencies of CD69^−^CD4^+^ and CD69^−^CD8^+^ T cells in tumor tissues (Figure 3B,C) showed lower recurrence rates.

Overall, these results underscore the importance of the local immune context and distinct prognostic roles of the T-cell subsets in liver and tumor compartments. The presence of CD69^−^ T cells—particularly CD4^+^ and CD8^+^ subsets—within the TME, is potentially impacting an effective anti-tumor response that improves RFS. In the liver tissues, CD69^+^CD103^+^CD8^+^ T cells may be essential to maintain immune surveillance and prevent recurrence post-hepatectomy.

### 3.4. Expressions of Exhaustion and Cytotoxicity Markers of Each T-Cell Subpopulation Within Tumor and Background Liver

Finally, we investigated PD-1, which is an exhaustion marker that is upregulated by chronic antigenic stimulation, and NKG2D, which is related to the antigen-independent cytotoxicity in each subpopulation. Representative histograms (Figure 4A) and quantification (Figure 4B) revealed that PD-1 expression was generally higher in CD69^+^ T cells than in CD69^−^ T cells. This difference was particularly pronounced in tumor tissues, suggesting that a more exhausted phenotype was associated with CD69^+^ T cells. The presence of CD69^−^CD4^+^ and CD69^−^CD8^+^ T cells, characterized by lower PD-1 expression, was associated with better prognosis, indicating that reduced exhausted functional CD69^−^ T-cell population may contribute to improved clinical outcomes. On the other hand, NKG2D also showed higher expressions on CD8^+^ T cells, particularly in CD69^+^CD103^+^CD8^+^ T cells, in both tissue types. Considering that CD69^+^CD103^+^CD8^+^ T cells in the liver and tumor tissues influence RFS differently, their contributions to tumor immune surveillance and tumor progression may depend on their specific location. Alcohol-related etiology influenced PD-1 expression levels. PD-1 was uniformly expressed in CD69^+^ populations; however, its expression on CD69^−^CD4^+^ and CD69^−^CD8^+^ T cells in tumor tissues differed significantly between patients with and without alcohol etiology (Figure 5A). In contrast, NKG2D expression patterns were less influenced by alcohol intake.

Furthermore, in tumor tissues, a significant negative correlation was observed between the frequency of CD69^−^CD4^+^ and CD69^−^CD8^+^ T cells and PD-1 MFI (Figure 5B). This finding aligns with the observation that CD69^−^ T cells exhibit relatively low levels of PD-1 expression and are less prone to exhaustion. Notably, the inverse correlation between the frequency of CD69^−^ T cells and PD-1 expression suggests that this subset maintains more effective functional activity. These results support the idea that CD69^−^ T cells play a crucial role in sustaining anti-tumor immune responses within the TME and ultimately contribute to improved clinical outcomes.

## 4. Discussion

In this study, we investigated the prognostic implications of distinct T-cell subsets infiltrating both the liver and tumor tissues of patients undergoing surgical resection for HCC. Our key findings reveal that the relative distribution and phenotypic characteristics of these subsets have distinct effects on RFS. Specifically, within the TME, a higher frequency of CD69^−^ T cells—comprising the CD4^+^ and CD8^+^ lineages—significantly improved RFS. However, the presence of CD69^+^CD103^+^CD8^+^ T cells correlated closely with favorable prognosis in the liver compartments, suggesting that this population residing in the tissue may contribute to effective immune surveillance post-surgery. Overall, these findings underscore the functional heterogeneity of liver-infiltrating T cells and highlight the need for a more detailed understanding of immune cell dynamics in HCC.

Generally, increased tumor-infiltrating T cells were associated with improved patient outcomes in various cancers, including HCC, in previous studies [22,23,24]. However, the TME in HCC is known to be influenced by chronic inflammation and cirrhosis, creating a complex and often immunosuppressive niche that can reprogram immune cell populations [25,26]. The presence of tumor-infiltrating T cells often implies a more robust anti-tumor immune response; however, the functional diversity among these subsets can yield opposing prognostic implications [27,28]. For example, in certain studies, CD69^+^ and CD103^+^ T cells were associated with enhanced cytotoxicity and better outcomes [29,30,31]; contrastingly, others have demonstrated that under chronic antigenic stimulation and within immunosuppressive TMEs, these same markers may be associated with T-cell exhaustion and diminished anti-tumor efficacy [17,32,33,34].

Our findings help resolve this complexity by revealing that CD69^−^ T cells within the tumor—a subset typically considered to be circulating or residing less in tissues—were strongly associated with improved RFS. These CD69^−^ T cells exhibited lower levels of PD-1, indicating they were functionally less exhausted compared with CD69^+^ T cells. Additionally, reduced PD-1 expression on CD69^−^ T cells suggests that they may retain more robust effector functions, which contributes to sustained anti-tumor immunity. Accordingly, a reduced PD-1 expression on CD69^−^ populations correlated with an increased frequency of these cells, reinforcing the concept that CD69^−^ T cells serve as a relatively “functional” subset capable of effectively contributing to anti-tumor response.

PD-1 is generally considered a marker of T-cell exhaustion, but its co-expression with CD69—particularly in T_RM_ cells—may reflect tissue retention and adaptation rather than exhaustion. In our study, CD69^+^ T cells showed higher PD-1 expression than CD69^−^ cells, while CD38 levels, a marker of recent activation, were comparable between the groups (Appendix A). This suggests that CD69^+^PD-1^+^ T cells likely represent tissue-adapted T_RM_ cells rather than exhausted or recently activated cells. This interpretation aligns with previous reports showing that T_RM_ cells can express PD-1 even in a non-activated state [35].

Notably, we identified CD69^+^CD103^+^CD8^+^ T cells as a favorable prognostic subset within the liver tissues. This observation implies that tissue-resident T cells situated in the broader liver parenchyma—beyond the immediate tumor border—may continuously survey and contain emerging malignant clones, thus preventing early recurrence [15,36,37,38,39]. Furthermore, CD69^+^ T_RM_ cells can become exhausted in a highly immunosuppressive TME [30,32,40]; however, their presence in non-tumor areas may allow them to function under less suppressive conditions, effectively serving as a frontline defense against microscopic residual disease occurring after resection [36,41].

Another notable finding was that those with alcohol-related etiology showed alterations in T-cell subset frequencies and higher PD-1 expression, particularly in CD69^−^ T cells. Alcohol consumption has been linked to immune dysregulation and T-cell dysfunction, potentially augmenting the immunosuppressive milieu [42,43,44]. These etiological factors may influence the functional states and distribution of tumor-infiltrating T cells, necessitating tailored immunotherapeutic strategies that will target the underlying liver disease etiology.

Nonetheless, this study has some limitations. We did not assess antigen specificity; hence, it is unclear whether CD69^−^ and CD69^+^ T cells are directed against tumor-specific antigens. Moreover, deeper functional and molecular characterization is warranted to delineate the exact mechanisms by which these subsets exert their effects on tumor surveillance and recurrence.

In addition, some variables demonstrated hazard ratios with lower bounds of the confidence intervals approaching 1.0, indicating borderline statistical significance. This may be attributed to limited sample sizes within specific subsets or skewed data distribution. Accordingly, further validation in larger, independent cohorts will be required to ensure the robustness and clinical relevance of these findings.

## 5. Conclusions

In conclusion, our findings highlight distinct prognostic roles for CD69^−^ and CD69^+^CD103^+^CD8^+^ T cells in HCC. The beneficial impact of CD69^−^ T cells within the tumor and CD69^+^CD103^+^CD8^+^ T cells in liver tissues underscores the complexity of T cell-mediated immunity in HCC and the importance of spatial context. These insights provide a rationale for developing individualized immunotherapeutic approaches that capitalize on the unique functional profiles and locations of these T-cell subsets, ultimately aiming to improve long-term outcomes for HCC patients.

## Figures and Tables

**Figure 1 cancers-17-01548-f001:**
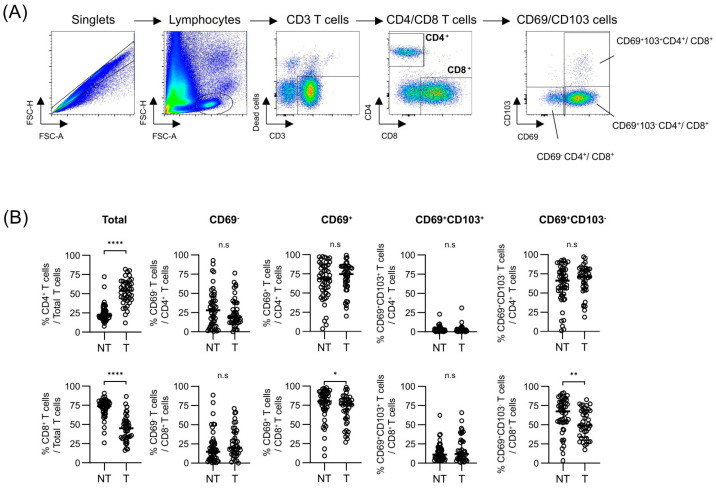
Flow cytometry analysis of CD69 and CD103 expression in CD4^+^ and CD8^+^ T cells in non-tumor liver and tumor tissues. (**A**) Gating strategy for identifying CD69 and CD103 subsets within CD4^+^ and CD8^+^ T cells. (**B**) Comparison of the relative frequencies and CD69/CD103 marker expression between NT and T. Statistical significance is indicated as follows: * *p* < 0.05, ** *p* < 0.01, and **** *p* < 0.0001; n.s., not significant. Statistical analysis was performed using the Mann–Whitney U test. n = 49 for NT, and n = 40 for T. NT, non-tumor tissue; T, tumor tissue.

**Figure 2 cancers-17-01548-f002:**
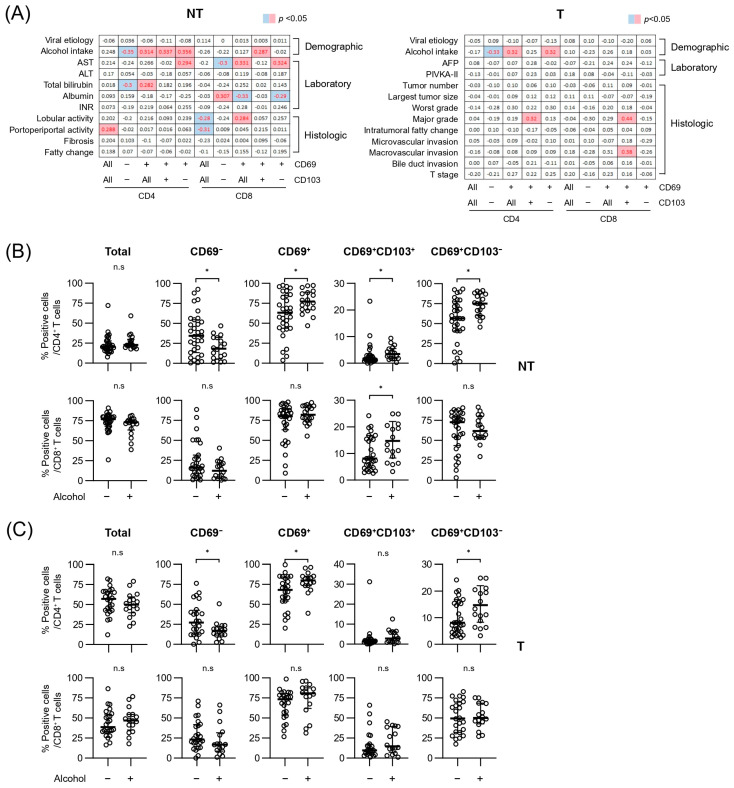
Correlations between T-cell subpopulations and clinicopathological factors in non-tumor and tumor tissues. (**A**) Spearman’s correlation analysis between T-cell subpopulations and clinicopathological factors in NT and T. Positive correlations are shown in red, and negative correlations are shown in blue, with statistical significance set at *p* < 0.05. (**B**,**C**) Comparison of the relative frequencies of T-cell subpopulations based on alcohol consumption in NT and T. In the NT group, non-alcohol (n = 31) and alcohol (n = 18) patients were analyzed; in the T group, non-alcohol (n = 24) and alcohol (n = 16) patients were analyzed. Statistical analysis was performed using the Mann–Whitney U test. Statistical significance is indicated as follows: * *p* < 0.05; n.s., not significant; NT, non-tumor tissue; T, tumor tissue.

**Figure 3 cancers-17-01548-f003:**
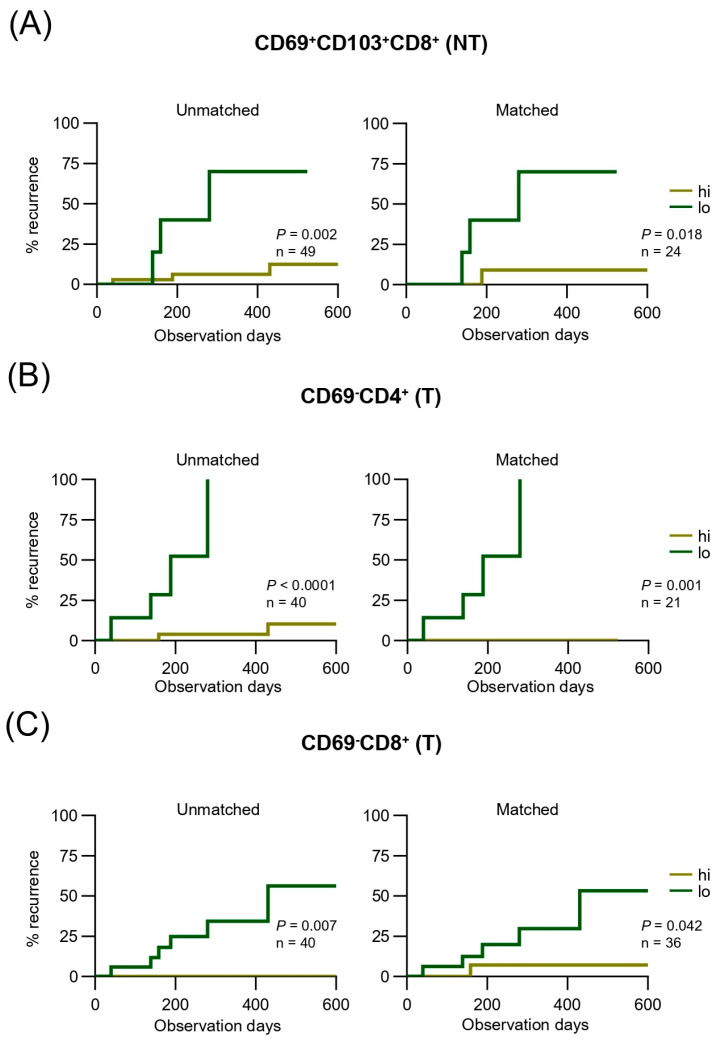
Kaplan–Meier survival analysis of T-cell subpopulations in non-tumor and tumor tissues. (**A**–**C**) The unmatched cohort includes all eligible patients without adjusting for confounding variables, providing a comprehensive view of survival outcomes. The matched cohort was adjusted for key clinicopathological variables, including age, sex, tumor stage, and etiology, ensuring balanced group comparisons and minimizing potential biases. (**A**) RFS curves comparing the frequencies of CD69^+^CD103^+^CD8^+^ T cells in NT. (**B**) RFS curves comparing the frequencies of CD69^−^CD4^+^ T cells in T. (**C**) RFS curves comparing the frequencies of CD69^−^CD8^+^ T cells in T. NT, non-tumor tissue; T, tumor tissue.

**Figure 4 cancers-17-01548-f004:**
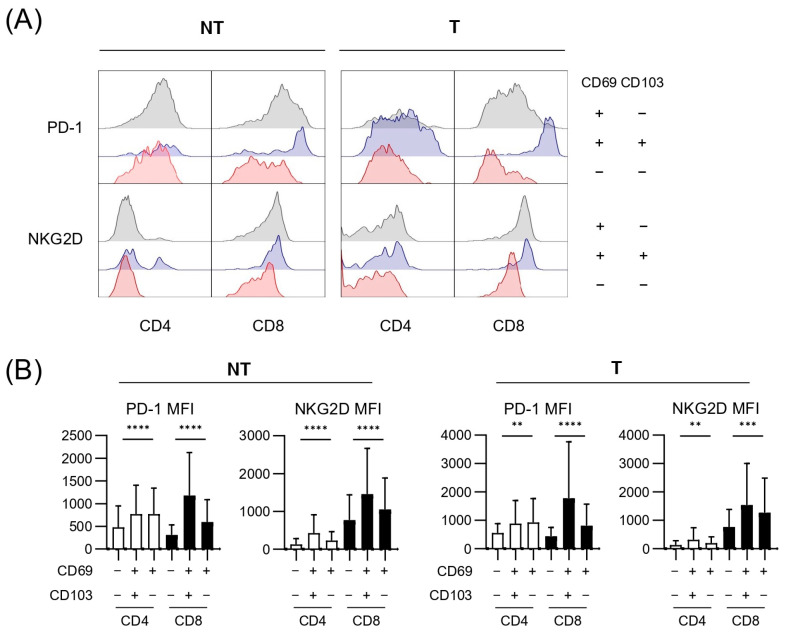
PD-1 and NKG2D expression levels in CD4^+^ and CD8^+^ T-cell subsets based on CD69 and CD103 expression in non-tumor and tumor tissues. (**A**) Representative histograms showing PD-1 and NKG2D expression in CD4^+^ and CD8^+^ T cells across CD69 and CD103 subsets in NT and T. (**B**) MFI of PD-1 and NKG2D in CD4^+^ and CD8^+^ T cells from NT and T. Analyses were performed on 32 NT samples and 24 T samples. Statistical analysis was performed using one-way analysis of variance. Statistical significance is indicated as follows: ** *p* < 0.01, *** *p* < 0.001, and **** *p* < 0.0001; n.s., not significant. NT, non-tumor tissue; T, tumor tissue; MFI, mean fluorescence intensity; PD-1, programmed death protein 1; NKG2D, natural killer group 2 member D.

**Figure 5 cancers-17-01548-f005:**
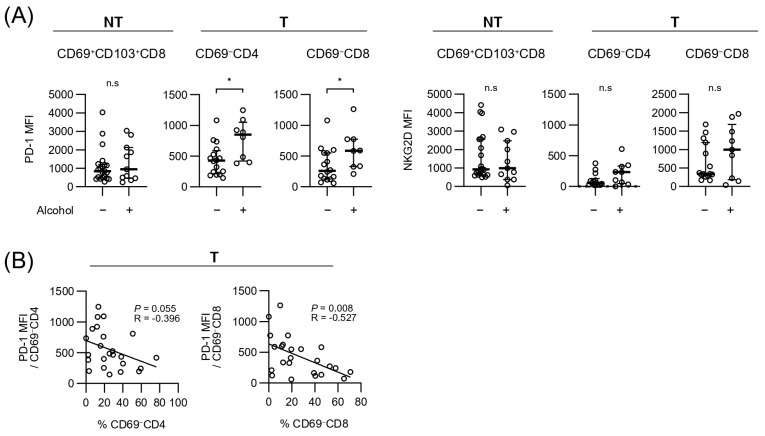
PD-1 and NKG2D expression levels in T-cell subsets based on alcohol etiology. (**A**) MFI of PD-1 and NKG2D in the CD69^+^CD103^+^CD8^+^ T-cell subset from NT and the CD69^−^CD4^+^ and CD69^−^CD8^+^ T-cell subsets from T, comparing alcohol and non-alcohol etiology groups. In the NT group, non-alcohol (n = 21) and alcohol (n = 11) patients were analyzed; in the T group, non-alcohol (n = 16) and alcohol (n = 9) patients were analyzed. Statistical analysis was performed using the Mann–Whitney U test. (**B**) Correlation between the percentage of CD69^−^CD4^+^ and CD69^−^CD8^+^ T cells and MFI of PD-1 in T (n = 24). Statistical significance (*p*-values) and correlation coefficients (R-values) are indicated in each panel. The Pearson correlation coefficient was applied to evaluate the correlation between the variables. Statistical significance is indicated as follows: * *p* < 0.05; n.s., not significant. NT, non-tumor tissue; T, tumor tissue; MFI, mean fluorescence intensity; PD-1, programmed death protein 1; NKG2D, natural killer group 2 member D.

**Table 1 cancers-17-01548-t001:** Baseline characteristics of enrolled patients.

	Total (n = 57)
Age (y)	62.7 ± 8.8
Male gender (n, %)	43 (75)
Etiology	
HBV (n, %)	28 (49)
Alcohol (n, %)	9 (16)
HBV + Alc (n, %)	12 (21)
MASH (n, %)	4 (7)
Unknown (n, %)	4 (7)
AST (U/L)	32.0 (23.0–39.0)
ALT (U/L)	29.0 (20.0–45.0)
Total bilirubin (mg/dL)	0.7 (0.5–1.0)
Albumin (g/dL)	4.4 (4.1–4.6)
AFP (ng/mL)	4.5 (2.7–47.5)
PIVKA-II (mAU/mL)	43.0 (20.0–274.0)
Tumor number	1.0 (1.0–1.0)
Largest tumor size	3.7 (2.0–5.6)
Worst tumor grade	3.0 (2.0–3.0)
Major tumor grade	2.0 (2.0–2.0)
Microvascular invasion (n, %)	19 (33)
Pathologic T stage	1.0 (1.0–2.0)

Data are presented as n (%) or mean ± standard deviation. Clinicopathological factors are presented as medians (interquartile range). AFP, alpha fetoprotein; AST, aspartate aminotransferase; ALT, alanine aminotransferase; HBV, hepatitis B virus; MASH, metabolic-associated steatohepatitis; PIVKA-II, protein induced by vitamin K absence or antagonist-II.

**Table 2 cancers-17-01548-t002:** Predictive value of tissue-infiltrating T-cell populations and clinicopathologic factors regarding recurrence-free survival using univariate Cox regression analysis.

Non-tumor/Tumor Tissue/ Clinicopathological	Population	HR (95% CI)	*p*	cutoff
NT	CD69^−^CD4	0.98 (0.93–1.02)	0.294	
NT	CD69^+^CD4	1.03 (0.98–1.07)	0.275	
NT	CD69^+^CD103^+^CD4	0.98 (0.89–1.08)	0.712	
NT	CD69^+^CD103^−^CD4	1.03 (0.98–1.09)	0.184	
NT	CD69^−^CD8	0.96 (0.89–1.03)	0.258	
**NT**	**CD69^+^CD8**	**1.97 (1.04–3.71)**	**0.037**	**85.6**
**NT**	**CD69^+^CD103^+^CD8**	**0.41 (0.23–0.73)**	**0.002**	**3.7**
**NT**	**CD69^+^CD103^−^CD8**	**2.14 (1.33–3.44)**	**<0.001**	**82.3**
**T**	**CD69^−^CD4**	**0.48 (0.32–0.72)**	**<0.001**	**11.5**
**T**	**CD69^+^CD4**	**2.23 (1.44–3.45)**	**<0.001**	**85.4**
**T**	**CD69^+^CD103^+^CD4**	**1.89 (1.19–3.01)**	**0.007**	**1.6**
**T**	**CD69^+^CD103^−^CD4**	**1.93 (1.22–3.05)**	**0.005**	**81.2**
**T**	**CD69^−^CD8**	**0.46 (0.26–0.81)**	**0.007**	**19.5**
**T**	**CD69^+^CD8**	**2.15 (1.33–3.49)**	**0.002**	**76.5**
**T**	**CD69^+^CD103^+^CD8**	**1.63 (1.01–2.65)**	**0.049**	**41.1**
**T**	**CD69^+^CD103^−^CD8**	**1.74 (1.04–2.93)**	**0.036**	**75.3**
Clinicopathological	Age	1.00 (0.97–1.03)	0.964	
Clinicopathological	Male gender	1.41 (0.46–4.30)	0.756	
Clinicopathological	Viral etiology	0.67 (0.31–1.42)	0.295	
Clinicopathological	Alcohol etiology	1.34 (0.27–6.66)	0.72	
Clinicopathological	AST	1.00 (0.99–1.01)	0.901	
Clinicopathological	ALT	0.99 (0.98–1.01)	0.473	
Clinicopathological	Total bilirubin	0.86 (0.31–2.34)	0.761	
Clinicopathological	Albumin	0.56 (0.28–1.13)	0.107	
Clinicopathological	INR	1.15 (0.09–14.76)	0.913	
**Clinicopathological**	**AFP**	**1.86 (1.18–2.93)**	**0.007**	**102**
**Clinicopathological**	**PIVKA-II**	**2.07 (1.26–3.39)**	**0.004**	**493**
Clinicopathological	Tumor number	1.19 (0.52–2.72)	0.688	
**Clinicopathological**	**Largest tumor size**	**2.82 (1.47–5.41)**	**0.002**	**6.9**
Clinicopathological	Worst tumor grade	1.45 (0.75–2.81)	0.264	
Clinicopathological	Major tumor grade	1.67 (0.80–3.49)	0.172	
Clinicopathological	microvascular invasion	1.55 (0.47–5.11)	0.469	
**Clinicopathological**	**Pathologic T stage**	**1.94 (1.22–3.08)**	**0.005**	**3**
Clinicopathological	Histologic liver fibrosis	1.18 (0.59–2.39)	0.64	

Bold values indicate statistical significance (*p* < 0.05). PSM, propensity-score matching; n.s, not-significant; NT, non-tumor; T, tumor; AST, aspartate aminotransferase; ALT, alanine aminotransferase; INR, international normalized ratio; AFP, alpha-fetoprotein; PIVKA-II, protein induced by vitamin K antagonist-II.

## Data Availability

All data relevant to the study are included in the article or uploaded as online Appendix A.

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
