# Peer review of "Differential Infiltration of T-Cell Populations in Tumor and Liver Tissues Predicts Recurrence-Free Survival in Surgically Resected Hepatocellular Carcinoma"

_cancers, 2025, doi:10.3390/cancers17091548_

Round 1
Reviewer 1 Report (Previous Reviewer 2)
Comments and Suggestions for Authors
Authors have addressed previous issues and this has improved the overall quality of the manuscript. This paper warrants publication in cancers.
Author Response
Comments 1 : Authors have addressed previous issues and this has improved the overall quality of the manuscript. This paper warrants publication in cancers.
Response 1 : We sincerely appreciate the reviewer’s support for the publication of our manuscript in Cancers.
Reviewer 2 Report (Previous Reviewer 3)
Comments and Suggestions for Authors
The authors now have re done their stat analysis re Table 2 where previously several HR crossed 1.0 and find now that thwery dont .The explanation for this change is inadequate .Please explain the differences in methodology to the editors and why they result in different HRs
Author Response
Please see the attachment.

This manuscript is a resubmission of an earlier submission. The following is a list of the peer review reports and author responses from that submission.
Round 1
Reviewer 1 Report
Comments and Suggestions for Authors
This study presents an interesting finding regarding CD69⁺ and CD69⁻ T cells in HCC and liver tissues, highlighting their distinct impact on RFS. The study provides strong evidence for assessing the subpopulations of CD69⁺/CD69⁻ T cells in relation to HCC prognosis. This paper lays a good foundation for future studies investigating the underlying mechanisms.
Overall, I find the paper well-organized and scientifically sound. The research is interesting, and the experiments appear to be well-planned and executed.
The study investigates the abundance patterns of CD69⁺ and CD69⁻ T cells in liver tissue and HCC, and explores their associations with clinical and histopathological parameters.
This study is both original and relevant. It offers a comprehensive analysis of CD69 expression patterns in T cells infiltrating HCC and normal liver tissue. This area is not well characterized, especially in relation to clinical outcomes, and the study addresses an important gap by exploring associations with patient clinical features.
The study provides a detailed profile of T cell subpopulations based on CD69 expression in the liver and TME. This is novel compared to previous work, which has mostly focused on tissue-resident memory T cells. Importantly, the authors also uncover new associations between CD69 expression patterns in specific T cell subsets and RFS, which is reported here for the first time.
The methodology is generally well-structured and logically presented.
The use of flow cytometry to profile T cell subsets, along with clinical data analysis and Kaplan–Meier curves, effectively supports the link between CD69 expression and RFS, addressing the study’s main objectives.
From my perspective, the manuscript is in good shape and is close to being ready for publication. Please feel free to reach out if you have any further questions or need additional clarification.
Author Response
Comments 1:
This study presents an interesting finding regarding CD69⁺ and CD69⁻ T cells in HCC and liver tissues, highlighting their distinct impact on RFS. The study provides strong evidence for assessing the subpopulations of CD69⁺/CD69⁻ T cells in relation to HCC prognosis. This paper lays a good foundation for future studies investigating the underlying mechanisms.
Overall, I find the paper well-organized and scientifically sound. The research is interesting, and the experiments appear to be well-planned and executed.
The study investigates the abundance patterns of CD69⁺ and CD69⁻ T cells in liver tissue and HCC, and explores their associations with clinical and histopathological parameters.
This study is both original and relevant. It offers a comprehensive analysis of CD69 expression patterns in T cells infiltrating HCC and normal liver tissue. This area is not well characterized, especially in relation to clinical outcomes, and the study addresses an important gap by exploring associations with patient clinical features.
The study provides a detailed profile of T cell subpopulations based on CD69 expression in the liver and TME. This is novel compared to previous work, which has mostly focused on tissue-resident memory T cells. Importantly, the authors also uncover new associations between CD69 expression patterns in specific T cell subsets and RFS, which is reported here for the first time.
The methodology is generally well-structured and logically presented.
The use of flow cytometry to profile T cell subsets, along with clinical data analysis and Kaplan–Meier curves, effectively supports the link between CD69 expression and RFS, addressing the study’s main objectives.
From my perspective, the manuscript is in good shape and is close to being ready for publication. Please feel free to reach out if you have any further questions or need additional clarification.
Response 1:
We sincerely appreciate the reviewer’s thoughtful and positive comments.
Reviewer 2 Report
Comments and Suggestions for Authors
Jang et al investigated the differences in the T cell infiltration in tumor and liver tissue in patients with resected hepatocellular carcinoma. So far, the immune infiltration in HCC has been unclear. In this study they found that CD3 expression did not affect overall survival. They found that higher frequency of CD69- in the TME showed improved recurrence-free survival. In liver tissue, increased frequencies of CD69+CD8+ and CD69+CD103-CD8+ T cells were associated with higher risk of recurrence.
These results add to the existing body of data on T cell infiltration in HCC. They could be used to inform patient care, as biomarkers and to predict response to treatment.
Paper is well written and figures are clear. Relevant and recent references have been included and the methods are clear with sufficient detail for reproduction by other researchers. Good discussion of limitations of study at the end of the discussion. This paper will provide a valuable contribution to cancers.
Questions:
Did the authors look at expression of any other exhaustion markers on the T cells, eg TIM-3, LAG-3
PD-1 is also an activation marker, is co-expression of both CD69 and PD-1 a sign of activation rather than exhaustion?
Minor Issues :
The authors need to add statistical tests and n values in all figure legends.
In methods section line 120 add in what enzymes are used
In methods section 123-126. Is there any difference in samples that have been frozen or not.
Table 2: why are some lines in bold
Reviewer 3 Report
Comments and Suggestions for Authors
The paper examines T cell subsets in human HCC tissues by cellular isolation and Flow cytometry .Note this is not Single cell RNA expression .
I have the following comments
- The alcohol effect needs t o be defined .There were only 9 p[patients with ALD but an additional 12 with HBV + alcohol .What was the definition of these two groups and are all 21 alcohol patients included in the figures when this was analysed ?
- There seems to be a contradiction in the statement:” tumour tissue contained a significantly higher number of CD8+ Tcells but lower number of CD4+ T cells “ ( 3.1 page 4 ) and Fig 1 B under total which seems to show the opposite .
- The correlates in Table 2 and Fig 3 with outcomes are a bit confusing. Generally if HRs crosses 1.10 in either direction then they are thought to be not significant but several so called significant differences cross 1.0 .This needs to be explained and addressed .The wide spread nature of the HRs need to be commented on .
- The results re PD-1 and NKG2D are of interest
- Where there any correlations with microvascular invasion and any T cell subsets ?
- If my concerns are correct then the discussion would need to be modified
Round 2
Reviewer 3 Report
Comments and Suggestions for Authors
Although there have been some improvements the issue of statistical differences when HRs cross 1 remains an issue
